# A Nano-Cleaning Fluid for Downhole Casing Cleaning

**DOI:** 10.3390/polym15061447

**Published:** 2023-03-14

**Authors:** Hanxuan Song, Yan Ye, Zhen Zhang, Shuang Wang, Tong Zhou, Jixiang Guo, Shiling Zhang

**Affiliations:** 1Unconventional Petroleum Research Institute, China University of Petroleum, Beijing 102200, China; 2State Key Laboratory of Petroleum Resource and Prospecting, China University of Petroleum, Beijing 102200, China; 3College of Petroleum Engineering, China University of Petroleum, Beijing 102200, China; 4Oil and Gas Engineering Research Institute of Tarim Oilfield, Korla 843300, China

**Keywords:** nano-emulsion, barium sulfate scale, polymers, compound evaluation system

## Abstract

In drilling and completion projects, sludge is formed as a byproduct when barite and oil are mixed, and later sticks to the casing. This phenomenon has caused a delay in drilling progress, and increased exploration and development costs. Since nano-emulsions have low interfacial surface tension, wetting, and reversal capabilities, this study used nano-emulsions with a particle size of about 14 nm to prepare a cleaning fluid system. This system enhances stability through the network structure in the fiber-reinforced system, and prepares a set of nano-cleaning fluids with adjustable density for ultra-deep wells. The effective viscosity of the nano-cleaning fluid reaches 11 mPa·s, and the system is stable for up to 8 h. In addition, this research independently developed an indoor evaluation instrument. Based on on-site parameters, the performance of the nano-cleaning fluid was evaluated from multiple angles by heating to 150 °C and pressurizing to 3.0 Mpa to simulate downhole temperature and pressure. The evaluation results show that the viscosity and shear value of the nano-cleaning fluid system is greatly affected by the fiber content, and the cleaning efficiency is greatly affected by the concentration of the nano-emulsion. Curve fitting shows that the average processing efficiency could reach 60–85% within 25 min and the cleaning efficiency has a linear relationship with time. The cleaning efficiency has a linear relationship with time, where R^2^ = 0.98335. The nano-cleaning fluid enables the deconstruction and carrying of the sludge attached to the well wall, which accomplishes the purpose of downhole cleaning.

## 1. Introduction

While the Tarim Basin is rich in oil and natural gas reserves [1,2,3], its role in China’s energy structure continues to grow and expand. Most of the wells are ultra-deep wells [4,5], and the lithology varies greatly from top to bottom in the Keshen block [6]. Some stratum fractures developed under the action of the complex pressure system [7]. A large number of wellbore instability problems occur during ultra-deep well drilling [8]. Oil-based drilling fluids [9,10] with the advantages of high-temperature resistance and strong inhibition are used to solve the problem of wellbore instability [11,12]. However, during the circulation of oil-based drilling fluids, and after the residual oil phase is mixed with barite, there is sedimentation [13]. Since the blockage is not soluble in acidic conditions [14,15], the conventional treatment method cannot be used to complete the cleaning. The accumulation of fouling elements harms extraction operations and raises the cost of exploration and development [16].

At present, there are various cleaning methods for underground oily sludge. From a technological point of view, some scholars use ultrasonic waves to decompose oily sludge [17,18,19]. However, during the on-site construction process, the chemical cleaning method is mainly used to clean the sludge. Some scholars achieve the decomposition of the sludge by chelating the solid phase of the sludge [20]. Zhifeng Luo [21] used diethylene triamine penta acetic acid (DTPA) and low-molecular compounds to prepare the chelating agent SA-20 to remove barium sulfate scaling, using physical crushing and chemical dissolution to treat barium sulfate scale effectively. However, most of the solid phase particles are encapsulated in the oil phase, and it is difficult for the chelating agent to make contact with it, so the full effect of the treatment cannot be achieved [22]. Therefore, the use of chemical agents to decompose organic matter in sludge is the most widely used treatment method. Hamidi Yasser [23] used an extraction method to treat oily dirt. The results show that chloroform has good performance as a solvent and can effectively extract hydrocarbons. Biao Mu [24] used liquid dimethyl ether to decompose oily sludge. Qinghua Bao [25] uses a binary mixed biosurfactant (rhamnolipid/sophorolipid, RL/SL) to strengthen the removing oil efficiency of oily sludge by thermal washing. Zhang Qi [26] proposed a novel method of combined degradation of oily sludge by surfactants with activated persulfate. The combined method significantly improved the degradation efficiency of oily sludge, and the removal rate reached 94.6±2.8%, and the oil content of the residual oily sludge was 0.57%, which reached the discharge standard. With the current research status, it is not difficult to see that most cleaning solutions have single components, low processing efficiency, and limited indoor evaluation methods. Most of them use a single method such as the weightlessness method or residual element detection method for evaluation [27], which cannot meet the requirements of efficient construction on site.

Nano-emulsions are fluids consisting of tiny emulsion structures containing an average radius of about 10–100 nm. During the emulsion formation process, surface active substances are adsorbed on the oil–water interface to form a stable interfacial film, which reduces the interfacial tension between oil and water. Nano-emulsions with ultra-low surface interfacial tension and strong cleaning ability have been widely used in oil repelling as well as oil washing in recent years. Ye Yan [28] applied nano-emulsions to oil sludge cleaning and achieved effective oil phase recovery. Mohammed K. Al-Sakkaf [12] used biosurfactants, crude oil, and water to prepare environmentally friendly nano-emulsions with good characteristics in EOR. Wang Jie [29] used the introduction of nano-emulsions in pressure fluid to achieve integrated pressure lift operation. The application of nano-emulsions in different fields of the oilfield also provides research ideas for this paper, based on the unique advantages of nano-emulsions, to introduce them into pipe-wall-pollution cleaning to achieve efficient cleaning of pipe walls and guarantee the smooth operation of oil and gas extraction.

This article analyzes the components of the obstruction taken from an existing ultra-deep well [30] using the principle of “deconstruction-carrying” and nano-emulsion as the basic liquid, and designs a set of high-density nano-cleaning liquids while drilling, as well as developing and conducting an indoor simulation of downhole oil-based blockages to provide guidance for on-site application practice.

## 2. Experimental Section

### 2.1. Materials

Chemicals used in nano-cleaning fluid preparation include Tween 80 (AR, 99.99%), n-pentane (AR. 99.99%), octane number (AR, 99.99%), fiber (PETROKMS. AR, 99.99%), and superfine calcium carbonate (particle size = 800 mesh, AR. 99.99%)

The downhole sludge used in the experiment was provided by Tarim Oilfield, and was used in oil–solid separation of oil sludge by Soxhlet extraction. As shown in Figure 1, the sludge in the picture is black cemented lumps. The Soxhlet extraction method is used to extract the oil phase of the sludge. The measured oil content of the sludge is 15.67% and the water content is 6.2%. Therefore, the obstruction is mainly based on the oil phase as the continuous phase to wrap and bond the solid phase, such as in barium sulfate. The formed cement is difficult to decompose and difficult to deform. Subsequently designed cleaning fluid needs to deconstruct the oil phase and solid phase, and then carry them to the ground through circulation.

The Varia EL Ⅲ type elemental analyzer produced by Elementar was used to analyze the element content of the inorganic solid phase substances in the oil sludge. The results are shown in Table 1. The solid phase component of the blockage in this block is mainly barium sulfate, accounting for about 75% of the blockage composition.

### 2.2. Preparation and Performance of Nano-Cleaning Fluid

#### 2.2.1. Preparation and Characterization of Nano-emulsions

Nano-emulsions have the advantages of low surface interfacial tension, small particle size, good stability, and strong biodegradability [27,28,31,32]. They have been well-applied in reservoir reconstruction, well drilling, and completion projects. Commonly used nano-emulsions can be prepared by mechanical energy input. Based on nano-emulsion, a downhole nano-cleaning fluid has been formulated to treat ultra-deep well obstructions, and an indoor evaluation has been carried out.

Use Tween 80 (or Tween 60) and n-butanol as the surfactant (S) and auxiliary surfactant (A) to construct the “S+A” system. Control the ratio of “S+A” and n-octane to 9:1, 8:2, 7:3, 6:4, 1:1, 4:6, 3:7, 2:8, and 1:9, followed by dropwise introduction of deionized water into a nano-emulsion system [33]. The pseudo-three-phase diagram of the obtained nano-emulsion is shown in Figure 2b.

As shown in Figure 2a, based on a given nano-emulsion system, add dispersed fibers to the system under high agitation conditions and stir for 10–15 min. Then, slowly add 600–800 mesh ultra-fine CaCO_3_ for weighting, and weigh zeta the nano-cleaning fluid system to 1.5–1.8g/cm^3^, after weighting and high stirring for 30 min to form a uniform nano-cleaning fluid. The nano-cleaning fluid formula is shown in Table 2.

#### 2.2.2. Nano-Emulsion Particle Size and Zeta Potential Test

A clear and transparent nano-emulsion liquid is prepared according to the three-phase diagram. Select nano-emulsion samples, using Manern’s Zetasizer 4 particle size analyzer and zeta potential measuring instrument to determine the nano-emulsion’s particle size and zeta potential.

#### 2.2.3. Surface Interfacial Tension of the Nano-Cleaning Fluid

A nano-cleaning fluid with a nano-emulsion concentration of 0.06–1.2% was prepared, and the surface tension of the cleaning fluid was tested by using an interfacial tensiometer (BZY-2).

#### 2.2.4. Wetting and Flipping Ability Test of P110 Steel by Nano-Cleaning Fluid

Contaminate the 5 cm × 1 cm P110 steel sheet with sludge (the material used for the casing), and then use the nano-cleaning fluid to soak the steel sheet, and use the contact angle analyzer (AY-PHb) to test the contact angle of the steel sheet before and after the nano-cleaning fluid soaking, to analyze the wetting and flipping effect of the nano-cleaning fluid on the casing.

#### 2.2.5. Sedimentation Stability Performance Test of Nano-Cleaning Fluid

The development of ultra-deep wells is faced with the problem of high downhole temperatures. The current cleaning fluid fails due to its poor temperature resistance. A vertical electric test bench (MIRCOE-300) is used to test the high-temperature stability of the nano-cleaning fluid kept at 180 °C for 8 h.

#### 2.2.6. Test of Stick–Cut Performance of Nano-Cleaning Fluid

By changing the nano-emulsion concentration in the formula, the amount of fiber added, and the density of the cleaning fluid, different nano-cleaning fluid formulations were formed, as shown in Table 3. Six-speed viscometer (ZNN-D6) was used to test the stick–cut properties of cleaning fluid with different formulations. These results can be used as guidance during future downhole applications.

### 2.3. Nano-Cleaning Fluid Cleaning Efficiency Test

The cleaning efficiency of the cleaning fluid was simulated by the cleaning efficiency evaluation instrument developed by the China University of Petroleum (Beijing). The device diagram is shown in Figure 3.

The cleaning efficiency evaluation instrument is composed of a cleaning liquid storage tank, a heatable simulated casing, a circulating pump, and a particle size analyzer. The sludge is heated at 150 °C and adsorbed on the simulated casing, and a circulating pump is used to circulate the cleaning liquid to flush the simulated casing. There is a screen at the bottom of the casing. When the sludge is dispersed into the cleaning liquid, it can pass through the screen to communicate with the simulated casing. The cleaning efficiency of the nano-cleaning fluid was calculated by measuring the quality difference before and after the simulated casing was cleaned. The particle size change of the nano-cleaning fluid before and after the cleaning was measured by a particle size analyzer, and the cleaning efficiency of the nano-cleaning fluid was evaluated in combination with the color change of the nano-cleaning fluid during the cleaning process. The evaluation system model established by the evaluation device is shown in Figure 4.

## 3. Results and Discussion

### 3.1. Nano-Cleaning Fluid Performance

According to the steps shown in Figure 2, the nano-cleaning liquid system was prepared, and the performance parameters of the nano-cleaning liquid were obtained through the characterization of instruments and equipment, as shown in Figure 5.

The nano-emulsion is the core component of the cleaning fluid, and its stability and particle size are the guarantees for efficient cleaning by the nano-cleaning fluid. The zeta potential of the nano-emulsion was measured to be −31.4, indicating that the emulsion system has good stability. The particle size test result of the nano-emulsion is shown in Figure 5a, and the particle size D_50_ value is about 14.8 nm.

Figure 5b shows the surface tension and interfacial tension of the nano-cleaning fluid with nano-emulsion concentrations of 0.6%, 0.8%, 1%, and 1.2%. Due to the presence of highly dispersed nano-emulsion in the nano-cleaning fluid, the surface tension of the nano-cleaning fluid is about 36 mN/m, which leads to a decrease in the surface tension of the nano-cleaning fluid. After adding fibers, the surface tension of the cleaning fluid increases by 1~2 mN/m. The dotted line in the figure is the interfacial tension of the cleaning fluid. The interfacial tension of the nano-cleaning fluid varies from 7 to 13 mN/m, with low interfacial tension, effectively cleaning sludge.

The lipophilic-treated P110 steel sheet was immersed in a nano-cleaning fluid with a concentration of 0.8%, and then the contact angle was measured at different time intervals. After 30 min, the contact angle between the P110 steel sheet and the water droplet changes from 101° to 27°, the wettability changes from hydrophobic to hydrophilic, and the wetting reversal occurs. It is confirmed that the nano-cleaning fluid has efficient wetting and turning ability, and good cleaning ability.

Figure 5c shows the change in surface and interfacial tension of lipophilic-treated P110 steel after being soaked in 0.8% nano-cleaning fluid. After 30 min, the contact angle between the P110 steel sheet and the water droplet changes from 101° to 27°, the wettability changes from hydrophobic to hydrophilic, and the wetting reversal occurs. It is confirmed that the nano-cleaning fluid has efficient wetting and turning ability, and good cleaning ability.

Aiming at the problem that the nano-cleaning fluid often fails in the high-temperature downhole environment of ultra-deep wells, the high-temperature stability of the nano-cleaning fluid measured by the vertical motorized test stand (MIRCOE-300) is shown in Figure 5d. The resistance value received is between 0.15–0.20N and remains stable. The nano-cleaning fluid can maintain good stability, is beneficial for carrying downhole sludge, and does not affect circulation.

### 3.2. Analysis of Cleaning Performance of Nano-Cleaning Fluid

#### 3.2.1. Analysis of the Stick–Cut Performance of Nano-Cleaning Fluid

In the cleaning process, the rheological properties of the cleaning fluid have a certain influence on the treatment effect and fluid circulation, and the rheological parameters of the cleaning fluid are studied by using the measurement and release method of drilling fluid. Using nano-cleaning fluids of different formulas to clean the sludge, the viscosity and shear performance of the nano-cleaning fluid after the measurement is shown in Figure 6.

Figure 6a shows the apparent viscosity (AV) of the nano-cleaning fluid after cleaning the sludge, which means that under a given speed gradient, the quotient of the corresponding shear stress is divided by the shear rate. In the figure, the apparent viscosity is lower when the nano-emulsion concentration is lower and the fiber content is higher. When the fiber content is decreased, the apparent viscosity increases because the fiber entangles the ultrafine calcium particles and the oily sludge, which eventually increases the system’s density.

Figure 6b shows the change in plastic viscosity (PV) with the concentration of nano-emulsion and the amount of fiber added. The plastic viscosity reflects the dynamic equilibrium between the suspended solid particles and the solid strength of the internal friction between the particles and the liquid phase and inside the continuous liquid phase. At higher fiber loadings, the relative internal friction also increases.

Figure 6c depicts the dynamic shear force (YP), which represents the mutual attraction between the active solid particles in the cleaning slurry’s laminar flow state to produce the internal resistance measurement. Its value decreases as the concentration of the emulsion increases and increases as the number of fibers increases. Through the changing trend of the dynamic shear force of the viscosity, the optimal nano-emulsion concentration and the optimal fiber addition amount are screened out to calculate the cleaning efficiency. At 0.06%–0.5% nano-emulsion concentration and 0.9–1.1g fiber addition, the viscosity and shear value of the nano-cleaning fluid is the best, which can meet the requirements for downhole cleaning.

#### 3.2.2. Evaluation of the Cleaning Effect of Nano-Cleaning Fluid

According to the method in Figure 3, the stirring speed was 300 r/min, the oven temperature was 150 °C, and the duration was 25 min. The designed volume–mass ratio of nano-cleaning fluid to sludge is 35:1. Indoor evaluation and analysis are performed best under the above conditions. The changes in cleaning solution before and after cleaning are shown in Figure 7.

As can be seen from Figure 7, the cleaning solution has an efficient dispersion performance. As seen through the particle size analysis in Figure 7a for the cleaning solution particle size distribution, it contains only one peak, mainly for the ultra-fine calcium peak. After a period of processing, cleaning particle size analysis, as shown in Figure 7b, shows that the particle size distribution at this time contains two peaks: the original ultra-fine calcium peak and, after the dissolution of oil-containing matter, the peak of barium sulfate scale. At this time, the oil-bearing barium sulfate scale was dispersed into the cleaning slurry, and the overall particle size was about 20 μm, which could be smoothly returned to the well with the slurry.

Figure 8 is a photo of the cleaning slurry for different cleaning time periods, and shows how quickly the nano-cleaning fluid’s color changes while it is being processed. These are the color images of five samples in ascending order (samples 1–5 in the bottom picture from left to right) taken at various processing times, from 0, 5, 10, 15, and 25 min, respectively. It can be seen from the photos that as the treatment time increases, the color of the nano-cleaning fluid darkens, which means that the amount of sludge dissolved in the nano-cleaning fluid is gradually increasing. In this cleaning system, the main role of the nano-emulsion is to disintegrate the oil sludge structure, stripping it of oil components, so that the solid phase particles wrapped in oil sludge disperse. Fiber provides the system with a mesh structure, so that the dispersed particles are suspended in the system, and the cleaning fluid circulation ensures a smooth return to the ground.

Figure 9 shows the observation photos of the oil sludge solid phase in different cleaning stages. From the figure, it can be seen that the oil sludge solid phase reaches a good dispersion under the action of cleaning liquid and entangles with fiber to a certain extent, which can be effectively suspended in the cleaning slurry. The mechanism of action of the cleaning fluid can also be observed through the photos of the sludge at different stages, confirming that the cleaning fluid has an efficient cleaning and carrying effect.

#### 3.2.3. Intuitive Phenomena during the Cleaning Process

The different formulas of the nano-cleaning fluids are shown in Table 3, and m_0_ and m_1_ are weighed according to the evaluation method in Figure 3. The cleaning efficiency is calculated according to the given formula ($\eta =\frac{m_{1}-m_{0}}{m_{0}}$). The calculated fitting results are shown in Figure 10.

As shown in Figure 10, by comparing the cleaning efficiency, it is found that the concentration of the nano-emulsion has a greater impact on the treatment efficiency. Moreover, after three-dimensional fitting, the cleaning efficiency is greatly affected by the nano-emulsion, and the cleaning efficiency can reach 74.885% in 25 min. A good cleaning effect is achieved.

#### 3.2.4. Research on the Cleaning Mechanism of Nano-Cleaning Fluid

The cleaning mechanism of nano-cleaning fluid was studied by fitting the experimental data, and its action mechanism was described. The fitting results are shown in Figure 11.

Figure 11 is a graph of the change in cleaning efficiency over time. The nano-cleaning fluid of 350 mL with a density of 1.51 g/cm^3^, a concentration of 1% nano-emulsion, and a fiber addition of 0.8 g was simulated indoors. The cleaning efficiency was tested for different periods within a 25 min range and then curve fitting was performed. The cleaning efficiency changes linearly with time, the fitting curve is y = 3.66 + 1.6x, and the fitting height is R^2^ = 0.98335. According to the analysis based on the principle, the nano-cleaning fluid mainly has a physical dispersion effect in the auction process, so it does not cause secondary pollution and can meet the requirements of efficient treatment. After the cleaning, the sludge can be taken out of circulation.

## 4. Conclusions

The percentages of oil, water, and barium sulfate in the downhole oil sludge of the Tarim deep block are 15.67%, 6.2%, and 75%, respectively. The sludge is well-cemented and not easily soluble in acid, which seriously affects the development progress. The nano-emulsion prepared in this study has an interfacial tension of about 4.5 mN/m and a surface tension of 35 mN/m. It can change the wetting angle of the oil sludge surface from 101° to 27° and has a strong wetting reversal ability. A set of nano-cleaning solutions with adjustable density were developed based on nano-emulsion and through a fiber-reinforced system network structure. The apparent viscosity of the nano-cleaning fluid is 11mPa-s and the stabilization time can reach 8 h, which can be used for the cleaning treatment of downhole oil mud in the field. During the treatment of oil mud, the particle size changes from a single peak to a bimodal peak, and deconstruction and dispersion of sludge exists, and the average treatment efficiency could reach 74.885%. The nano-emulsion in the nano-cleaning fluid disperses the oil sludge solid phase to realize oil–solid separation, and the separated solid phase is wound by the fiber to hinder its settlement, and carried out of the ground through the circulation system. The change in treatment volume with time is obtained by curve fitting, and the curve y = 3.66 + 1.6x is highly fitted with R^2^ = 0.98335, which confirms that the treatment mechanism is mainly “deconstruction-carry.“ The nano-cleaning solution with adjustable density developed in this study has high treatment efficiency and can be used for ultra-deep well cleaning. In addition, the indoor simulation method also guides field application, and provides a guarantee for the realization of clean and environmentally friendly production.

## Figures and Tables

**Figure 1 polymers-15-01447-f001:**
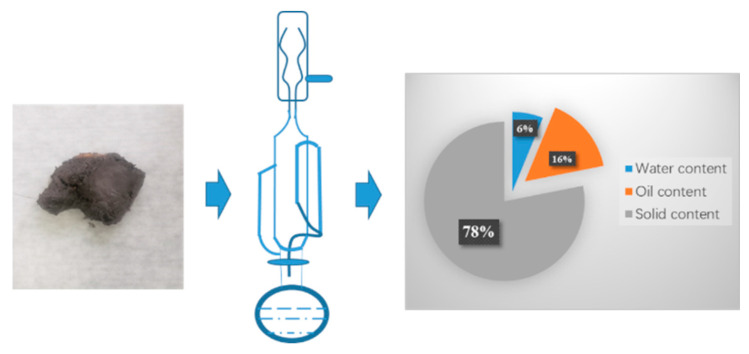
Soxhlet extraction method to measure oil and water content.

**Figure 2 polymers-15-01447-f002:**
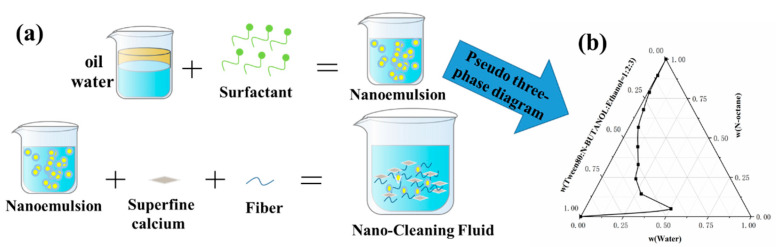
(**a**) Nanoemulsion preparation flow; (**b**) Nanoemulsion pseudo-triple-phase diagram.

**Figure 3 polymers-15-01447-f003:**
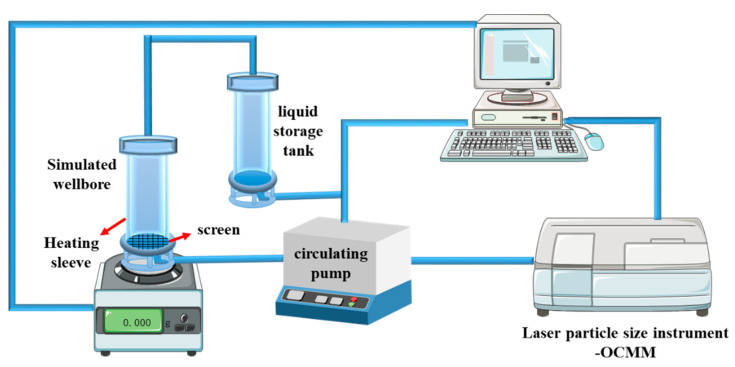
Nano-cleaning fluid evaluation device.

**Figure 4 polymers-15-01447-f004:**
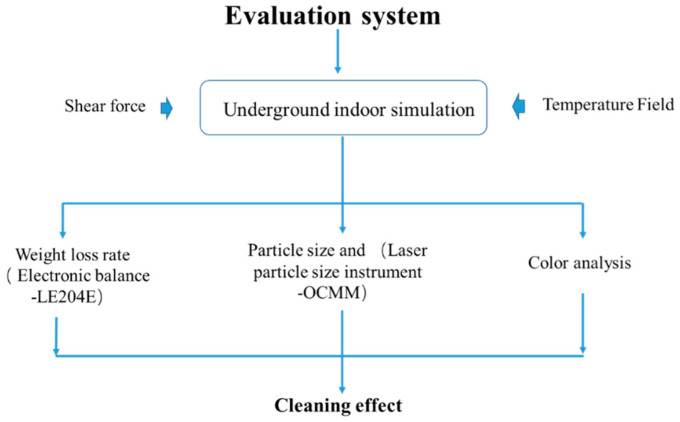
Cleaning fluid evaluation model.

**Figure 5 polymers-15-01447-f005:**
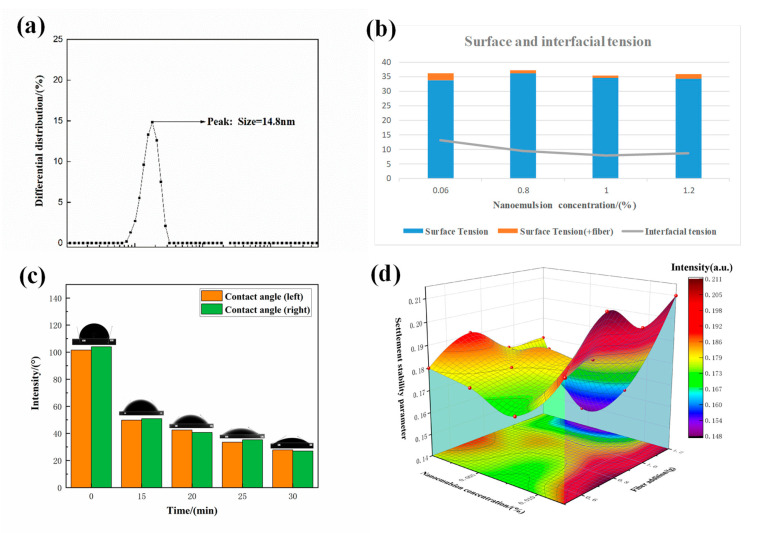
(**a**) Nano-emulsion particle size distribution; (**b**) surface tension of nano-cleaning fluid; (**c**) nano-emulsion wetting reversal performance test; (**d**) stable settlement of nano-cleaning fluid.

**Figure 6 polymers-15-01447-f006:**
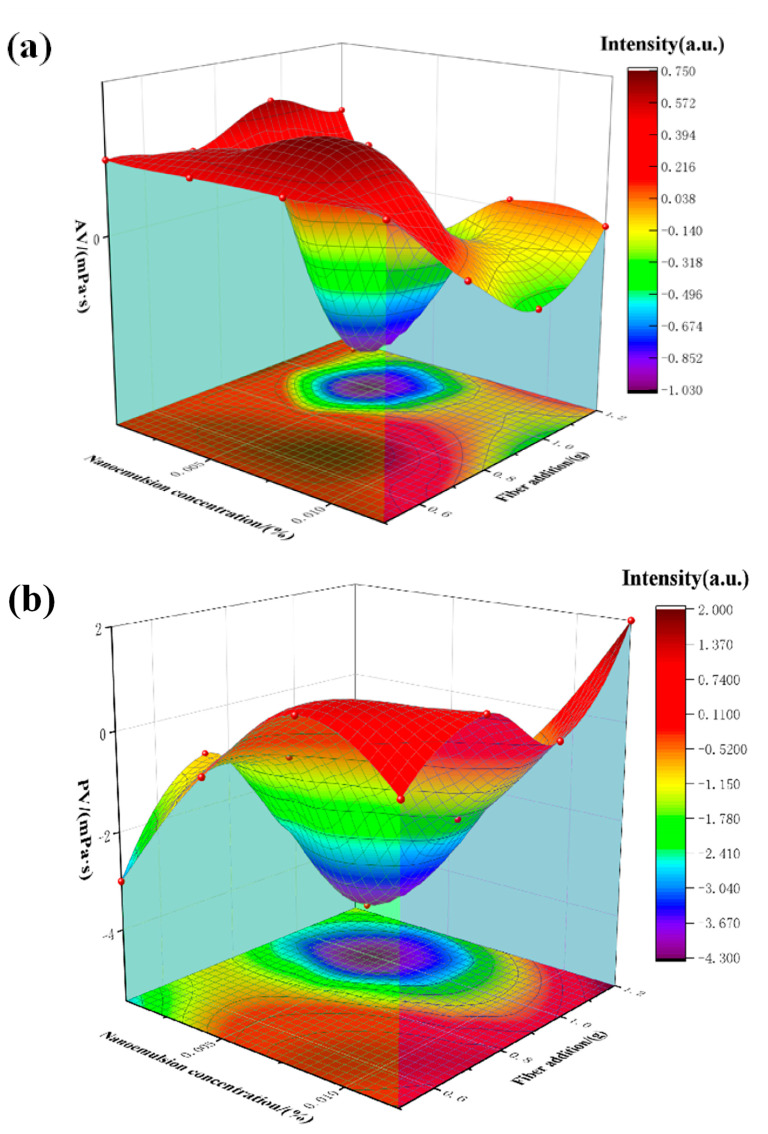
(**a**) Influence model of each component addition on apparent viscosity; (**b**) influence model of each component addition on plastic viscosity; (**c**) influence model of each component addition on yield value.

**Figure 7 polymers-15-01447-f007:**
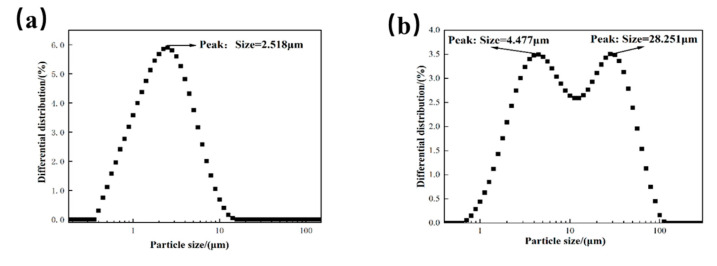
(**a**) Cleaning slurry particle size, (**b**) particle size of cleaning slurry and sludge mixture.

**Figure 8 polymers-15-01447-f008:**
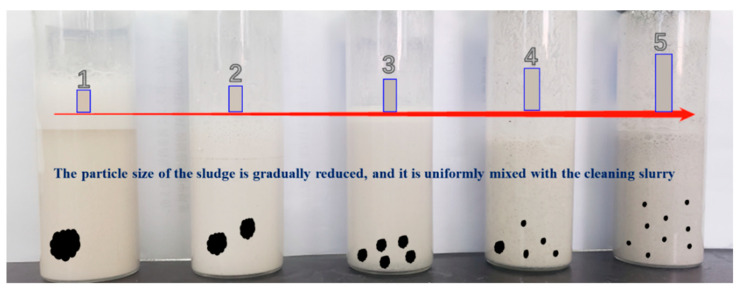
Color change in nano-cleaning fluid with processing time.

**Figure 9 polymers-15-01447-f009:**
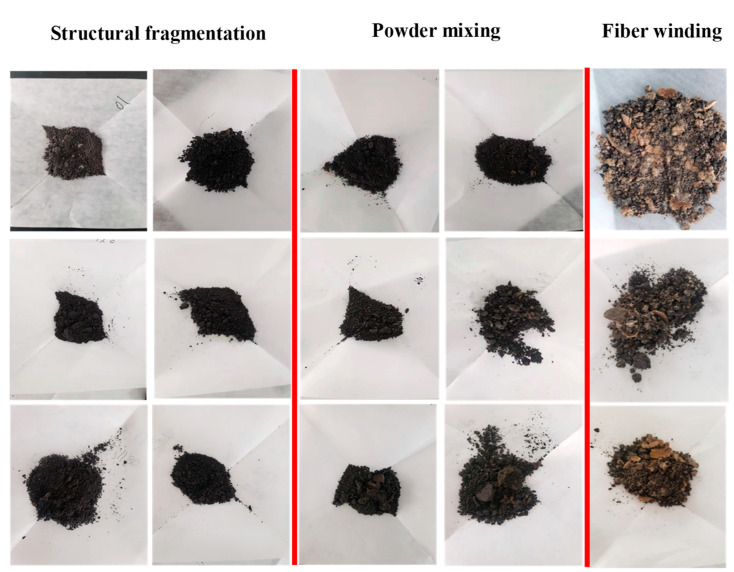
Solid phase at different cleaning stages.

**Figure 10 polymers-15-01447-f010:**
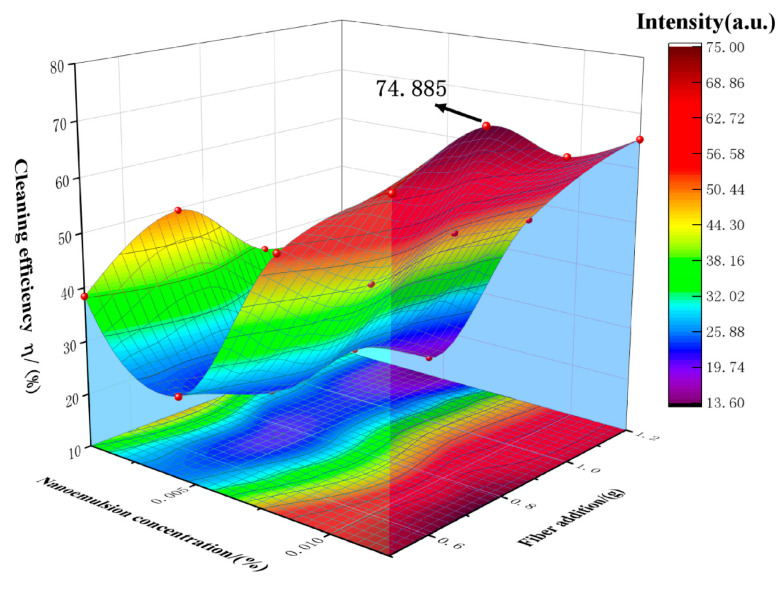
Nano-cleaning fluid cleaning efficiency.

**Figure 11 polymers-15-01447-f011:**
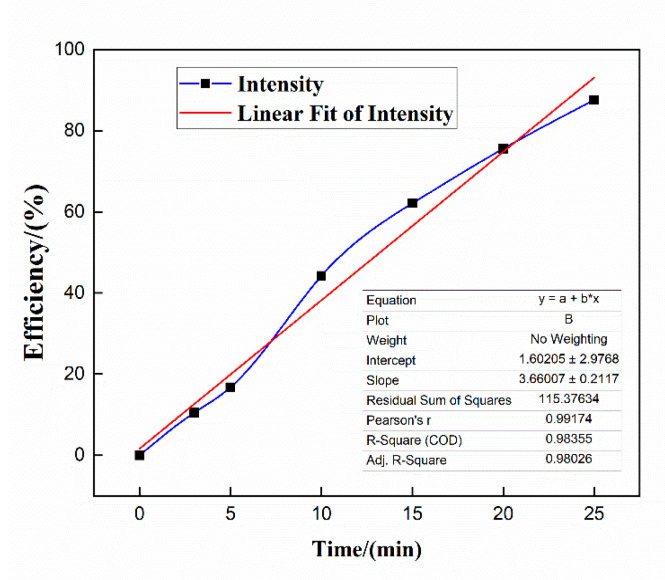
Fitting curve of cleaning efficiency.

**Table 1 polymers-15-01447-t001:** Analysis of solid phase elements in oil sludge.

Element	Ba	S	Ca	Fe	Si	Mg	Zn	Mn	Al	Na	Other
Content %	51.999	25.067	7.353	4.869	3.628	2.387	1.36	1.005	0.773	0.409	1.15

**Table 2 polymers-15-01447-t002:** Nano-cleaning fluid formula.

Species	Tween 80 (or Tween 60)	N-Butanol	N-Octane	Fiber	Barite Perform
Proportion	2.1–5.6%	0.6–1.6%	0.3–0.8%	0.09–0.23%	200%

**Table 3 polymers-15-01447-t003:** Experimental program of nano-cleaning fluid.

Number	Nano-Emulsion Concentration/(%)	Fiber Addition/(g)	Density/(g/cm^3^)
1	0.06	0.5	1.5
2	0.06	0.8	1.5
3	0.06	1	1.5
4	0.06	1.2	1.5
5	0.8	0.5	1.5
6	0.8	0.8	1.5
7	0.8	1	1.5
8	0.8	1.2	1.5
9	1	0.5	1.5
10	1	0.8	1.5
11	1	1	1.5
12	1	1.2	1.5
13	1.2	0.5	1.5
14	1.2	0.8	1.5
15	1.2	1	1.5
16	1.2	1.2	1.5

## Data Availability

Not applicable.

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
