# Peer review of "A Nano-Cleaning Fluid for Downhole Casing Cleaning"

_polymers, 2023, doi:10.3390/polym15061447_

Round 1

Reviewer 1 Report

In this study, the authors tried to evaluate the performance of the nano-cleaning fluid for downhole casing cleaning. The work is significant that can be accepted for publication after considering the following comments: 

1. Which type of nanoparticles is used? It is not obvious. 

2. Why the authors called new nanofluid? It is better to remove the word "new". 

3. The authors mentioned that ultra-low IFT can be achieved by nanos. But this can't be seen in their IFT results shown in Figure 6b. Why? 

4. On Figure 6c, it should be left angle not lift angle, correct it. 

5. It would be much better if the authors can show some pictures of the prepared nanofluids, especially for the stability observation through the glass vessels. 

6. The authors can get benefit from the following article: 

https://www.sciencedirect.com/science/article/pii/S0920410520302084

7. Self-description is needed for some Figures. i.e. 6, 9 and 11. 

Reviewer 2 Report

"A new nano-cleaning fluid for downhole casing cleaning" is an original paper dealing with self-cleaning nanoemulsions for drilling. The paper is well organized and quite well written. Some minor amendments can be useful:

1) Please make sure and check all style, grammar, and other issues. Also, you have some problems with the formatting. E.g. in the conclusion section, your equation "jumped"

2) Can you estimate the cost efficiency of your emulsion? I think it can be quite straightforward and easy to do. The cost of this emulsion will be most important when drilling engineer should decide to use it of not

3) Can you propose a few possible areas for your future field trial? In other words, from your point of view, where this technology can be approbated first?
